# The Chronic Diseases Clinic of Ifakara (CDCI)—Establishing a Model Clinic for Chronic Care Delivery in Rural Sub-Saharan Africa

**DOI:** 10.3390/diseases10040072

**Published:** 2022-09-30

**Authors:** Maja Weisser, Martin Rohacek, Robert Ndege, Ezekiel Luoga, Andrew Katende, Getrud J. Mollel, Winfrid Gingo, Fiona Vanobberghen, Daniel H. Paris, Christoph Hatz, Manuel Battegay

**Affiliations:** 1Division of Infectious Diseases and Hospital Epidemiology, University Hospital Basel, 4031 Basel, Switzerland; 2University of Basel, 4001 Basel, Switzerland; 3Swiss Tropical and Public Health Institute, 4123 Allschwil, Switzerland; 4Ifakara Health Institute, Ifakara P.O. Box 53, Tanzania; 5St. Francis Referral Hospital, Ifakara P.O. Box 73, Tanzania

**Keywords:** chronic care, rural sub-Saharan Africa, HIV, Tuberculosis, non-communicable diseases

## Abstract

The rollout of antiretroviral drugs in sub-Saharan Africa to address the huge health impact of the HIV pandemic has been one of the largest projects undertaken in medical history and is an unprecedented medical success story. However, the path has been and still is characterized by many far reaching implementational challenges. Here, we report on the building and maintaining of a role model clinic in Ifakara, rural Southwestern Tanzania, within a collaborative project to support HIV services within the national program, training for staff and integrated research to better understand local needs and improve patients’ outcomes.

## 1. Introduction

Since the late eighties, the HIV pandemic has affected the African continent most tragically and disproportionally compared to other parts of the world. HIV/AIDS remains a major threat to the population. Eastern and Southern Africa continue to have the highest burden of people living with HIV (PLHIV)—20.6 million or 54% in 2021—despite tremendous progress in the reduction of new HIV infections by 61% since 2010 [1]. Adolescent girls remain at highest risk for new HIV infections with three times higher risks compared to males of the same age group.

According to the UNAIDS report 2022, in Tanzania, the HIV/AIDS incidence has further decreased to 0.96/100,000 (54,000 new infections in 2021). The number of AIDS-related deaths has halved since 2010 but remains unacceptably high with 29,000 people dying from AIDS in 2021. Currently, 88% of people know their HIV status (1st 95% of the UNAIDS cascade goals); 86% of PLHIV are in care (2nd 95%) and 83% are virally suppressed (3rd 95%) [1].

The implementation of integrated HIV care programs in rural settings remains challenging due to lack of trained staff and availability of specialized services [2,3,4]. Here, we describe a collaborative project to build up one of the first rural clinics to deliver HIV care services in support of the National AIDS Control Program of Tanzania, integration of mother and child care, TB services and services for non-communicable diseases as a model clinic for chronic disease care.

## 2. The HIV Pandemic in 2000—Implementation of a Rural Care and Treatment Center

After the Durban AIDS conference in 2000 with recognition of the huge gap in HIV programs in Africa, international collaborations started to tackle the HIV pandemic. In 2003 the U.S. President’s Emergency Plan for AIDS Relief (PEPFAR) was funded as the most important funding agency to implement HIV programs on a large scale in sub-Saharan Africa. While this allowed financing of the drug supply, implementation of HIV care programs remained challenging due to lack of trained healthcare staff, logistics and a healthcare system not equipped for chronic care. In 2003, during this period, the Chronic Diseases Clinic of Ifakara (CDCI) in rural Tanzania was founded as a joint collaboration between the St. Francis Referral Hospital in Ifakara, the Ifakara Health Institute in Tanzania, the Swiss Tropical and Public Health Institute and the University Hospital Basel in Switzerland. From the very beginning the concept was to establish a setting with preventive and clinical services at its center, and training and clinical research as equally important pillars to ensure the highest quality of care and sustainability through capacity building of staff and analysis of patients’ outcomes and specific needs of this rural area.

After a long period where only counselling of PLHIV was possible and 1.5 years of organizational work, building up clinical and laboratory structures and teams, the first patient was able to be treated on 1 May 2005. Since then, the CDCI has uninterruptedly delivered services to patients of Ifakara and the surrounding districts in the Kilombero Valley, serving as a referral site for very sick patients for a growing catchment area of now more than 1 million inhabitants.

## 3. Structure and Procedures of the Clinic

The focus of the CDCI is on HIV testing, enrolment into care, screening for opportunistic infections, and same-day initiation of antiretroviral treatment (“test and treat”). CDCI provides care for in- and outpatients with an HIV and/or tuberculosis infection according to the Tanzanian guidelines, collaborating closely with local implementing partners of national HIV programs, district and national governmental authorities (Figure 1 and Figure 2). Key elements are integration of the CDCI into the outpatient department of the hospital to reduce stigmatization, close links with other services and specialists. Furthermore, the establishment of an electronic patient data system with standardized electronic questionnaires (Open Medical Record System, www.openmrs.org) was central to allow simultaneous access for all collaborators in charge of patient services (clinicians, triage, registration, pharmacy, laboratory) contributing to a highly efficient workflow and systematic documentation for clinical and research purposes [2]. The system is continuously updated according to guideline changes and research necessities by a local statistics team with support for training and development from the partners.

Currently, forty staff members employed at the CDCI by the different partners ensure continuous care for about 4200 patients under active care in the outpatient department and inpatients with HIV and/or tuberculosis. Nurses triage outpatients and direct them towards clinical consultations at least once yearly, or in the presence of new symptoms, to the pharmacist for drug refill or to counselling services if indicated. The viral suppression rate for those attending the CDCI is high at 92% [3]. Medical doctors are trained to assess patients with complex medical conditions and supervise clinical officers. Professional and lay counsellors support patients with psychosocial health issues, poor adherence to drugs, stigmatization and challenges to care for HIV-infected dependents.

To reduce attrition, we offer a bundle of interventions for newly diagnosed patients aiming to reduce stigmatization in a structured program, namely (a) one-to-one counselling by an HIV-positive lay person, (b) viewing of a locally made video, in which PLHIV talk about their experience, (c) offering group therapy during waiting times provided by HIV-positive lay counsellors and (d) assessment of stigma and depression during the first year [4].

## 4. Kilombero and Ulanga Antiretroviral Cohort (KIULARCO)

The KIULARCO is one of the biggest and longest standing cohorts of PLHIV in East Africa with more than 12,000 consenting PLHIV enrolled since 2005 (Figure 3). The long-term nature of the cohort—currently 17 years—and the large set of systematic clinical and laboratory data collected together with a large biobank make this cohort unique [2,5]. Based on its data and biobank important scientific manuscripts on treatment outcomes [3,6,7], mother to child transmission [8,9,10], comorbidities [11,12,13,14,15,16,17], mortality [5,6] and other aspects were published and helped to adjust HIV service delivery to better meet patients’ needs.

To ensure high-quality care, immunologic and virologic monitoring was established at an early time-point and allowed the site to become the referral hub in viral load and early infant DNA PCR testing for the four surrounding districts in the Kilombero valley, closely collaborating with the national AIDS control program. Resistance testing has been established for patients failing on antiretroviral treatment, which is central in the management of patients and helps to inform stakeholders of the national antiretroviral programs [3,18,19].

## 5. Integration of Services

Screening, diagnosis and management of tuberculosis—the most common comorbidity in HIV-infected patients—has been fully integrated in the CDCI. All patients are screened according to the Tuberculosis and Leprosy National Control Program (NTLP) guidelines. Implementation of screening questions, chest X-ray and real-time PCR (Xpert MTB/RIF) [20] led to an increase in case detection. Additionally, we have established sonography for assessment of extrapulmonary tuberculosis (EPTB) [21] (Figure 4 and Figure 5).

## 6. ‘One Stop Clinic’: Integration of Antenatal and Under-Five Clinic

A successful part of the clinic is a family-centered program, the “One Stop Clinic”, which was implemented in 2013 to improve services for HIV-infected pregnant and breast-feeding mothers together with their partners and children. Services are offered by one team to the whole family under the same roof, located at the Reproductive and Child Care Clinic of the hospital. With this program, mother-to-child transmission has reduced to below 2% [8,9] with no transmission among virally suppressed breastfeeding women [10]. Age-appropriate disclosure services are offered to children living with HIV and psychological support to adolescents living with HIV is done by a trained counsellor until transition to adult HIV services is reached at the age of 19 years.

## 7. Non-Communicable Diseases

During long-term care of PLHIV, increasingly other comorbidities are recognized, importantly non-communicable diseases (NCDs) such as arterial hypertension [11], and other cardiovascular diseases, as well as cancer and chronic anaemia [12]. In 2021 a new heart and lung disease clinic was established in close proximity to the CDCI and the One Stop Clinic, sponsored by the Else Kröner Fresenius Foundation. This clinic offers specialized and chronic care for patients with heart and lung diseases such as, e.g., hypertensive heart diseases, valvular heart diseases, asthma, post-tuberculosis lung diseases and other pulmonary diseases. An important aspect is the creation of awareness on the importance of NCDs, which is addressed by regular outreach programs to the community, offering education, screening and referral for those with detected health issues.

## 8. Training

Every day, the first working hour is reserved for education and training of staff, including clinical case presentations, state of the art lectures on HIV and associated diseases, interdisciplinary discussion on patients with a high HIV viral load or sequencing results, and a journal club. Sessions are coordinated by a team member on a rotational basis, contributing to a continuous medical education and fostering clinical discussion among the team. Additionally, the CDCI aims to support trainings for Master and PhD programs and medical specialization.

## 9. Research

Many important research questions have already been answered through use of the prospective database of the KIULARCO Cohort [2,5] and it continues to provide a rich resource to address newly arising questions. Additionally, CDCI serves increasingly as a platform for clinical trials nested within the collaborative activities, such as a randomized controlled two center trial to study the added value of sonography in the management of patients with suspected extrapulmonary tuberculosis [22], the CoArtHA trial to identify the best antihypertensive drug strategy [23] or the GIVE-MOVE trial to analyze the impact of drug resistance testing in children and adolescents failing on first line treatment on the treatment outcome [24].

## 10. Conclusions and Perspectives

The platform for high-quality care, training and research is based on a longstanding and excellent collaboration between the key partners. A strong collaboration with national stakeholders enabled the CDCI to become a scalable model clinic for service delivery in rural sites. With the progress of antiretroviral drugs, “treatment as prevention” with “U=U”, i.e., “Undetectable equals Untransmittable”, care of utmost quality becomes increasingly important for patients and their beloved ones and will guide the future. The essence of the CDCI—good care for patients—continues to be the basis to tackle new developments, unforeseen challenges and to improve lives of people living with HIV.

## Figures and Tables

**Figure 1 diseases-10-00072-f001:**
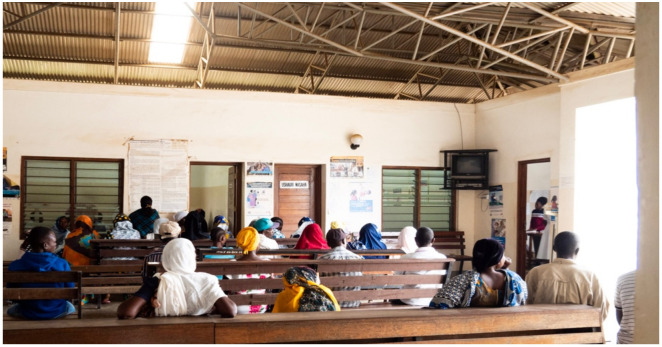
Waiting area of the Chronic Diseases Clinic of Ifakara.

**Figure 2 diseases-10-00072-f002:**
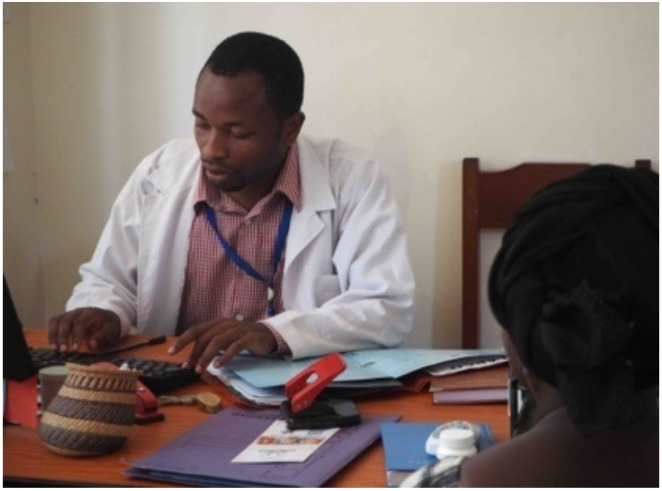
Consultation at the CDCI, Dr Ezekiel Luoga (head of CDCI, co-author).

**Figure 3 diseases-10-00072-f003:**
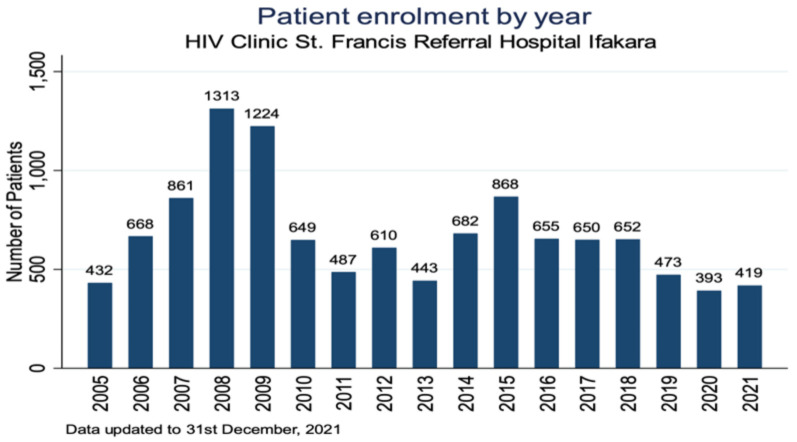
Yearly numbers of patients enrolled into the Kilombero and Ulanga Antiretroviral Cohort (KIULARCO) since 2005.5. Laboratory.

**Figure 4 diseases-10-00072-f004:**
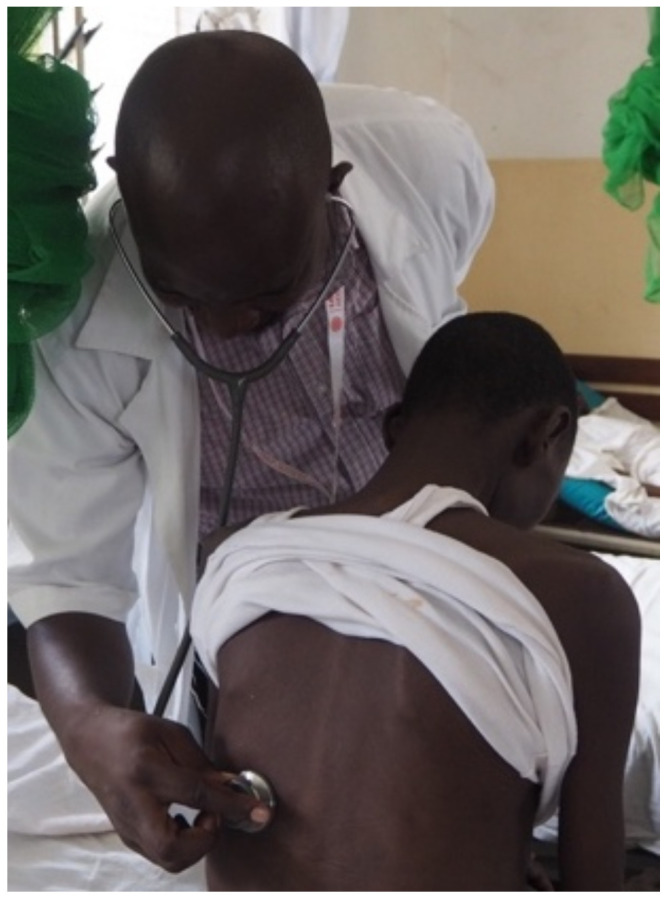
Clinical evaluation of a patient in the wards.

**Figure 5 diseases-10-00072-f005:**
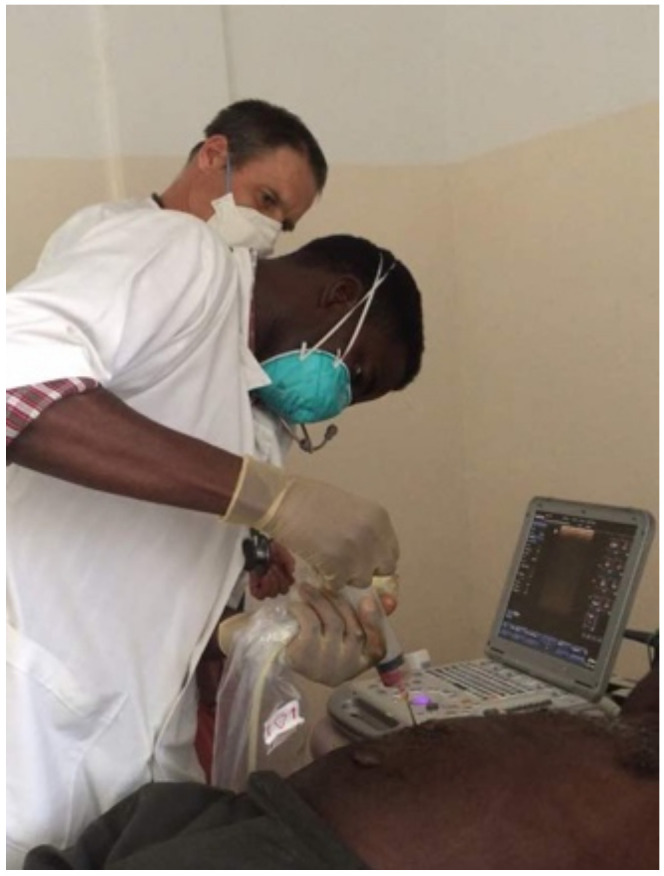
Ultrasound-guided puncture of a liver abscess.

## Data Availability

Not applicable.

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
