# Peer review of "The Chronic Diseases Clinic of Ifakara (CDCI)—Establishing a Model Clinic for Chronic Care Delivery in Rural Sub-Saharan Africa"

_diseases, 2022, doi:10.3390/diseases10040072_

Round 1

Reviewer 1 Report

The authors present in this short communication article a project for the foundation of a model clinic for chronic care delivery in rural sub-Saharan Africa. The article is of scientific imporatnce and may provide a model for the conduct of further activities in this region. 

I would personally like to see in a graph/table the actual expectations based on a potential statistical model and estimated by introducing the results of previous studies in the field. Patient expectations and drop-out rates may be commented as well. Relevant citations to the above include but are not limited to:

de-Graft Aikins A, Boynton P, Atanga LL. Developing effective chronic disease interventions in Africa: insights from Ghana and Cameroon. Global Health. 2010 Apr 19;6:6. doi: 10.1186/1744-8603-6-6. PMID: 20403170; PMCID: PMC2873935. Bischoff A, Ekoe T, Perone N, Slama S, Loutan L. Chronic disease management in Sub-Saharan Africa: whose business is it? Int J Environ Res Public Health. 2009 Aug;6(8):2258-70. doi: 10.3390/ijerph6082258. Epub 2009 Aug 14. PMID: 19742159; PMCID: PMC2738886. Ameh S, Klipstein-Grobusch K, D'ambruoso L, Kahn K, Tollman SM, Gómez-Olivé FX. Quality of integrated chronic disease care in rural South Africa: user and provider perspectives. Health Policy Plan. 2017 Mar 1;32(2):257-266. doi: 10.1093/heapol/czw118. PMID: 28207046; PMCID: PMC5400067. Ameh S, Klipstein-Grobusch K, D'ambruoso L, Kahn K, Tollman SM, Gómez-Olivé FX. Quality of integrated chronic disease care in rural South Africa: user and provider perspectives. Health Policy Plan. 2017 Mar 1;32(2):257-266. doi: 10.1093/heapol/czw118. PMID: 28207046; PMCID: PMC5400067.

Author Response

We thank the reviewer for this comment and suggestions. For the graph/table we however believe it is challenging to show meaningful numbers for an article like this (e.g, comparators for the treatment cascade using the UNAIDS 95-95-95), as this report is more a qualitative report on our settings.

We have added the above cited articles as requested in the Introduction

Reviewer 2 Report

Thank you for the possibility to review the manuscript entitled: "cxThe Chronic Diseases Clinic of Ifakara (CDCI) – establishing a 2 model clinic for chronic care delivery in rural sub-Saharan Africa": it is more interesting to read and very corious in its content.

However, the manuscript could be implemented in its contents, in the introduction and laso in its methodology and relating results: in this version it could be not classified as an Article.

Author Response

We agree this is not a scientific article but a project report. It is the journals’ decision whether this fits into the portfolio or not.

Reviewer 3 Report

This is an excellent description of a role model clinic. 

I do hope it is a scalable model 

A cost effective study would make this one stronger

Author Response

We thank for this observation. We have commented on scalability of the project in the conclusion part. While we agree with the importance of a cost-effectiveness study, this is not the content of this article and needs a separate work.

Round 2

Reviewer 2 Report

I think that tha manuscript will be processed further.